# OpenReview forum: "Local Linearity of LLMs Enables Activation Steering via Model-Based Linear Optimal Control"
_ICML.cc/2026/Conference — ICML 2026 regular_

### Official Review · Reviewer_ApYn · 2026-03-08

**Soundness:** 3
**Presentation:** 3
**Significance:** 3
**Originality:** 3
**Overall Recommendation:** 5
**Confidence:** 4

**Summary:**

This work developed a scalable, model-based optimal control framework for activation steering using online feedback. It also provided a theoretical analysis of A-LQR. Extensive experimental results on multiple LLMs demonstrated the good performance of the proposed method.

**Compliance With Llm Reviewing Policy:**

Affirmed.

**Final Justification:**

The idea of online steering is very interesting.

**Key Questions For Authors:**

See the comments above

**Limitations:**

See the comments above

**Strengths And Weaknesses:**

Strength:

1. It is a novel idea to perform activation steering using online feedback. Compared to existing works on offline steering, the online feedback is very interesting and promising.

2. It also offered a theoretical analysis of the error bound of the closed-loop trajectory tracking error.

3. Evaluation on multiple LLMs illustrated the good performance of the proposed method.

Weakness:
1. It is not clear about the computational costs of activation steering using online feedback. It would be great to report the computational costs.

2. Compared to more baselines. The proposed method should be compared with recent works on control-based activation steering, such as ReControl and ODESteer [1,2].

3. It may need to evaluate the transferability of this method on cross-datasets.

4. It would be great to study the effect of important parameters like \delta in the control theory on model performance.

References:
[1] ODESteer: A Unified ODE-Based Steering Framework for LLM Alignment, ICLR, 2026

[2] Kong et al. Aligning large language models with representation editing: A control perspective. Advances in Neural Information Processing Systems, 37, 2025.

---

> ### Author Rebuttal · Authors · 2026-03-31
>
> We sincerely thank the reviewer for the insightful comments. We have carefully addressed your comments below. Please let us know if you have any additional questions or concerns.
> [Link to supplemental materials](https://anonymous.4open.science/r/paper_supp-E476/_ICML_2026__Rebuttal_Figures_and_Tables_sanitized.pdf )
>
> ### **Q4.1: It is not clear about the computational costs of activation steering using online feedback. It would be great to report the computational costs.**
>
> We evaluate online runtime, including benchmarking against ODESteer. Our method incurs higher cost due to online feedback, requiring a d × d matrix multiplied by a d × 1 error vector. On an RTX 4090, A-LQR and S-PID reduce tokens-per-second by about 20%. However, this translates to under a 1 second increase for mid-length generations (~100–150 tokens), which we find acceptable for general LLM applications. Runtime remains competitive with ODESteer while outperforming all baselines on steering metrics. Notably, the small gap between A-LQR and S-PID suggests that the cost incurred by LQR is *minimal*, as PID is a canonically lightweight controller. Although A-LQR involves more complex operations than S-PID, GPU parallelization largely eliminates this difference. More results are in the supplemental.
>
> |Model|Original|A-LQR|S-PID|ODESteer|ActAdd|
> |---|---|---|---|---|---|
> |Llama-3.2-1B|95.04 ± 2.265|73.41 ± 0.3776|69.77 ± 2.0544|70.78 ± 0.2225|95.96 ± 0.3983|
> |Qwen-2.5-3B|43.64 ± 0.8994|32.93 ± 0.9404|32.90 ± 0.7902|37.05 ± 0.9763|43.88 ± 0.6184|
> |Gemma-2-9B|31.89 ± 0.5412|24.44 ± 0.4107|24.70 ± 0.6000|28.18 ± 0.3325|31.93 ± 0.3522|
>
>
> ### **Q4.2: Compare to more baselines.**
>
> We have added ODESteer as a baseline for toxicity mitigation, truthfulness, and runtime.
>
> Our method outperforms ODESteer on toxicity mitigation without any additional tuning. For truthfulness, ODESteer initially outperformed our original reported values for some models; however, with further tuning of the LQR controller, our method matches or surpasses these results. This sensitivity to parameters is a limitation we will discuss in a dedicated limitations section (see Reviewer YJUu). The updated values for truthfulness are summarized in the table below, and the results for toxicity are summarized at the supplemental.
>
> |Model|Method|T·I (↑)|True (%) (↑)|Info (%) (↑)|MMLU (%) (↑)|
> |---|---|---|---|---|---|
> |Gemma-2-2B|Original|48.64 ± 1.14|50.62 ± 1.28|96.08 ± 0.77|51.94 ± 1.63|
> ||ODESteer|63.50 ± 0.36|66.24 ± 0.77|95.86 ± 0.86|52.92 ± 1.75|
> ||S-PID|58.26 ± 1.12|62.06 ± 1.05|93.88 ± 0.88|50.78 ± 1.54|
> ||A-LQR|**67.81 ± 0.38**|73.17 ± 0.28|92.68 ± 0.47|48.76 ± 2.61|
> |Qwen-2.5-3B|Original|41.60 ± 1.23|43.28 ± 1.28|96.11 ± 0.18|67.84 ± 1.21|
> ||ODESteer|60.03 ± 1.32|64.01 ± 1.74|93.78 ± 0.55|65.14 ± 1.24|
> ||S-PID|52.11 ± 0.96|54.54 ± 0.90|95.54 ± 0.20|63.62 ± 0.91|
> ||A-LQR|**60.60 ± 1.49**|65.09 ± 1.08|93.10 ± 1.11|65.38 ± 1.07|
> |Qwen-2.5-14B|Original|53.17 ± 0.52|55.15 ± 0.52|96.40 ± 0.22|78.80 ± 0.00|
> ||ODESteer|71.56 ± 1.26|75.03 ± 1.12|95.37 ± 0.67|78.08 ± 0.77|
> ||S-PID|61.48 ± 0.87|63.55 ± 0.82|96.74 ± 0.59|75.90 ± 1.13|
> ||A-LQR|**76.28 ± 0.87**|80.81 ± 0.60|94.39 ± 0.54|71.66 ± 1.44|
>
>
> ### **Q4.3: Evaluate the transferability of this method on cross-datasets.**
>
> We conducted toxicity mitigation experiments on the Jigsaw Toxic Comment dataset [1] using A-LQR, while keeping the contrastive feature directions from the RealToxicityPrompts dataset. As summarized in the table below, our method transfers effectively to this new dataset. These results will be included in the appendix of the revised manuscript.
>
> [1] cjadams et al., “Toxic Comment…” 2017. Kaggl
>
> |Model|Setting|Toxicity (%) (↓)|PPL (↓)|
> |---|---|---|---|
> |Gemma-2B|Original|7.10 ± 0.86|17.88 ± 0.15|
> ||A-LQR|0.18 ± 0.12|19.66 ± 0.15|
> |Qwen-3B|Original|5.92 ± 0.49|17.20 ± 0.42|
> ||A-LQR|0.58 ± 0.19|19.47 ± 0.33|
> |Llama-8B|Original|7.18 ± 0.37|16.86 ± 0.92|
> ||A-LQR|0.24 ± 0.19|17.43 ± 0.44|
>
>
> ### **Q4.4: It would be great to study the effect of important parameters in the control theory on model performance.**
>
> To clarify, δz_k denotes the deviation between a state and its nominal, i.e., δz_k = z_k - z̄_k, and is not a tunable parameter. Other LQR parameters, such as the Q and R matrices, are ablated in Appendix D in the context of toxicity mitigation. Consistent with standard LQR intuition, increasing state costs Q yields more aggressive steering and greater toxicity reduction at the expense of output quality, while increasing control-effort costs R reduces toxicity less but preserves output quality.

---

> > ### Author Rebuttal · Reviewer_ApYn · 2026-04-01
> >
> > The authors have addressed my main concerns, thus I would like to raise the score.

---

> > > ### Author Response · Authors · 2026-04-08
> > >
> > > We sincerely thank the reviewer for their thoughtful review, and we are glad that we were able to address their main concerns.

---

### Official Review · Reviewer_abn9 · 2026-03-09

**Soundness:** 3
**Presentation:** 3
**Significance:** 2
**Originality:** 3
**Overall Recommendation:** 4
**Confidence:** 3

**Summary:**

This paper proposes Activation-LQR (A-LQR), a control-theoretic activation steering method for large language models that models transformer inference dynamics as a locally linear time-varying system and applies Linear Quadratic Regulator (LQR) control to compute feedback interventions on hidden activations. The approach defines Linear Feature Setpoints (LFS) using contrastive activation differences and computes layer-wise steering signals that track these semantic targets during inference. A key assumption of the method is that transformer layers exhibit sufficient local linearity to justify the use of classical optimal control tools. To support this assumption, the authors provide empirical analyses suggesting alignment between layer-wise Jacobians across different activation trajectories. Experiments on concept induction, toxicity mitigation, and truthfulness tasks across several open-source models indicate that A-LQR can outperform several existing activation steering baselines. To conclude, the paper frames activation steering as a closed-loop optimal control problem over approximate transformer dynamics.

**Compliance With Llm Reviewing Policy:**

Affirmed.

**Final Justification:**

The authors have adequately addressed my concerns regarding computational overhead. Overall, I am inclined toward acceptance and believe this work will encourage further research on applying control theory to LLM control, particularly in activation steering.

**Key Questions For Authors:**

1. How robust is the observed Jacobian alignment across prompts from different domains or adversarial inputs?

2. What is the computational cost of computing Jacobians for models larger than those used in the experiments?

3. How sensitive is performance to the choice of the nominal trajectory used for linearization?

4. Can the authors provide a direct runtime comparison against methods such as Activation Addition or PID-based steering?

5. Can the controller simultaneously track multiple feature directions?

**Limitations:**

No. The limitations are only partially discussed. The authors should more explicitly discuss the strong reliance on local linear approximations and the computational overhead associated with Jacobian estimation.

**Strengths And Weaknesses:**

## Strengths

1. The paper proposes a principled perspective by modeling LLM inference as a dynamical system and applying optimal control techniques. This formulation provides a useful bridge between control theory and representation engineering.

2. The work attempts to provide tracking error bounds under linearization assumptions. While the guarantees rely on strong assumptions, providing theoretical analysis is a positive step relative to many heuristic steering methods.

3. Most existing activation steering methods operate in an open-loop manner, injecting fixed vectors into activations. In contrast, the proposed method uses feedback based on current activation error, which is conceptually appealing.

4. The experiments evaluate several models and tasks (toxicity mitigation, truthfulness, and concept induction), providing some evidence that the method can influence generation behavior while maintaining reasonable performance on auxiliary metrics.

---

## Weaknesses
1. The entire method depends on the assumption that transformer layer dynamics are locally linear across reachable activations. The empirical evidence provided (Jacobian alignment across prompts) is limited and does not convincingly demonstrate that this property holds broadly across diverse prompts, adversarial inputs, or larger models.

2. The method requires computing layer-wise Jacobians of transformer blocks. While the paper proposes computing controller gains offline, obtaining these Jacobians for large models could still be computationally expensive. The paper does not convincingly demonstrate scalability to large modern LLMs.

3. Claims about minimal computational overhead are unclear. The abstract claims the method introduces minimal computational overhead, yet the approach requires Jacobian computation and matrix multiplications at each layer. The paper does not provide clear runtime comparisons against existing steering approaches.

---

> ### Author Rebuttal · Authors · 2026-03-31
>
> We thank the reviewers for their thoughtful comments and feedback. We have reviewed them carefully and provide our rebuttal below. Please let us know if you have any additional questions or concerns.
> [Supplemental materials](https://anonymous.4open.science/r/paper_supp-E476/_ICML_2026__Rebuttal_Figures_and_Tables_sanitized.pdf )
>
> ### **Q3.1: The empirical linearity evidence provided is limited and does not demonstrate that it holds broadly**
> To clarify, our claim is that transformer layer dynamics along task-relevant trajectories can be approximated as locally linear, sufficient for feedback control. Empirically, Jacobians from related prompts align well, enabling a single linear controller to generalize. We extend alignment evaluations to additional domains (legal text, Python code) and a larger model (Qwen2.5-32B), showing consistent alignment (results in supplemental).
>
> While adversarial or highly out-of-distribution inputs may reduce alignment, the locally linear formulation enables linear time-varying robust control methods (e.g., SLS [1]) to mitigate such effects. We will clarify this assumption, expand evaluation, and discuss robustness in the revision.
>
> [1] Anderson et al. “System Level Synthesis.” arXiv:1904.01634
>
> ### **Q3.2: The paper does not convincingly demonstrate scalability to large modern LLMs**
> We perform supplemental experiments on Qwen2.5-32B and Llama-3-70B. Despite high offline memory requirements (see Reviewer BD7b), A-LQR scales effectively to these larger models. Notably, A-LQR performance is comparable to smaller models, while baselines degrade or scale less reliably (see supplemental).
>
> We also emphasize that, to the best of our knowledge, both the 32B and 70B models are larger than any model considered in prior activation steering literature.
>
> |Model|Method|CLS Tox. (%) (↓)|PPL (↓)|
> |---|---|---|---|
> |Qwen2.5-32B|Original|3.72±0.57|8.61±0.13|
> ||S-PID|0.28±0.15|8.86±0.06|
> ||A-LQR|**0.22±0.12**|8.83±0.10|
> |Llama3.1-70B|Original|5.64±1.19|7.65±0.08|
> ||S-PID|2.32±0.10|6.66±0.11|
> ||A-LQR|**1.12±0.41**|7.89±0.18|
>
>
> ### **Q3.3: How robust is the observed Jacobian alignment?**
> We conduct additional Jacobian alignment experiments with prompts from different domains (law, Python code), showing that alignment remains high across tasks (see supplemental). While alignment generally supports effective steering, adversarial or highly out-of-distribution prompts may violate local linearity. This limitation will be acknowledged in a dedicated limitations section (see reviewer YJUu).
>
>
> ### **Q3.4/5/6: Claims about minimal computational overhead are unclear**
> We summarize compute benchmarks, including offline memory (see Reviewer BD7b) and online runtime (Reviewer ApYn). While the method scales effectively, memory requirements for large models (32B, 70B) are high and exceed typical workstation capacities. Future work should explore optimizations, such as low-rank approximations of Jacobians or caching feedback matrices, to reduce this footprint. That said, most prior work focuses on 2–9B models, whose Jacobians fit a single 24GB VRAM workstation (e.g., RTX 4090).
>
>
> ### **Q3.7: How sensitive is performance to the choice of the nominal trajectory used for linearization?**
> We find empirically that steering performance is not very sensitive to the choice of nominal trajectory. As an example, we steer to induce “football” concept using a Jacobian corresponding to a “football” prompt, and a Jacobian corresponding to  “dog”. In this case, the "dog" Jacobian actually outperforms the "football" Jacobian, although this is likely due to stochasticity in the text generation.
> | λ   | dog Jacobian | football Jacobian |
> |-----|--------------|-----------------|
> | 0.5 | 0.22         | 0.16            |
> | 1.5 | 0.88         | 0.86            |
> | 2.5 | 0.93         | 0.82            |
>
>
> ### **Q3.8: Can the controller simultaneously track multiple feature directions?**
> Yes. The method can steer multiple concepts simultaneously, by modifying the pipeline to include different setpoints corresponding to different concepts (i.e., two different contrastive vectors, with two different online error signals track simultaneously). As an example, we demonstrate using A-LQR to generate outputs combining church-related and football-related content where both concepts share the same $\lambda$ value.
>
> | λ | football | church |
> |-----|---------|--------|
> | 0.5 | 0.14 | 0.02 |
> | 1 | 0.61 | 0.18 |
> | 1.5 | 0.81 | 0.27 |
> | 2 | 0.88 | 0.47 |
> | 2.5 | 0.90 | 0.62 |
>
>
> ### **Q3.9: The limitations are only partially discussed.**
> In the revision, we will add a dedicated section, which we outline in more detail in response to Reviewer YJUu. Specifically, we will discuss: (1) memory overhead of Jacobians, and (2) reliance on local linearity, which may degrade on adversarial inputs. However, the linear formulation enables the use of robust control tools to mitigate the effects of adversaries, such as SLS [1].

---

> > ### Author Rebuttal · Reviewer_abn9 · 2026-04-03
> >
> > Thank you for the detailed rebuttal and the additional experiments. I appreciate the effort to address the raised concerns.
> >
> > The new scalability results on Qwen2.5-32B and Llama3.1-70B are a meaningful addition, and the nominal trajectory sensitivity experiment (Q3.7) and multi-concept steering demonstration (Q3.8) directly and convincingly address those questions.
> >
> > However, I maintain my concerns on the following points:
> >
> > 1. **OOD/adversarial robustness (Q3.3):** Extending alignment evaluations to law and Python code domains is a welcome addition, but does not address the core concern, i.e. the behavior of Jacobian alignment and tracking error bounds under adversarial or significantly out-of-distribution inputs. The pointer to SLS-based robust control as a mitigation strategy is a direction for future work, not a characterization of the current method's guarantees in these regimes.
> >
> > 2. **Computational overhead (Q3.4/5/6):** The rebuttal acknowledges that memory requirements for 32B and 70B models exceed typical workstation capacities. This confirms rather than resolves the scalability concern. A practical method should either fit within reasonable hardware budgets or provide a clear path (e.g., low-rank Jacobian approximations) with demonstrated results.
> >
> >
> > I recognize the paper makes a principled contribution and the empirical results are broadly strong. I am open to revising my score. For now, I maintain my score of 3 (Weak Reject).

---

> > > ### Author Response · Authors · 2026-04-08
> > >
> > > We thank the reviewer for their thoughtful reply. We address the follow-up concerns below. ([Supplemental](https://anonymous.4open.science/r/paper_supp-E476/reply.pdf))
> > >
> > > **1\.** We clarify that our framework does not provide formal guarantees under adversarial or strongly out of distribution inputs. Rather, Thm. 4.2 states that, given a Lipschitz bound on the linearization remainder, the tracking error follows the prescribed bound. Should an input be adversarially selected such that the remainder violates this assumption, the bound may not hold. If calibrated on such a remainder, the theorem still applies, but may yield a less informative (e.g., looser) bound as this remainder increases.
> > >
> > > To evaluate this regime, we stress test both of the linearization experiments (Sec. 5) with OOD and adversarial inputs. We interpret OOD inputs as naturally shifted prompts (e.g., cross-language prompts, steering Japanese into English), and adversarial prompts as jailbreak-style prompts intended to elicit harmful behaviors.
> > >
> > > For the tracking experiments, we find that the empirical worst-case bound becomes looser under these prompts (due to larger linearization error). However, the true empirical tracking performance remains stable, despite the bound becoming less informative.
> > >
> > > Similarly, we augment the alignment evaluations to include heterogeneous datasets, including OOD or adversarial prompts. We observe that the alignment within datasets exhibit stronger alignment, but there remains nontrivial alignment between datasets, consistent with Fig. 5(b) (see supplemental). For example, the heterogeneous alignments for Qwen-2.5-3B (0.23-0.39) remain above thresholds identified in response to Reviewer YJUu (>0.2) which indicate effective steering.
> > >
> > > We therefore propose revising the paper to make our claim about local linearity more precise. In particular, we observe that A-LQR is empirically resilient to natural input shifts, but we do not provide formal guarantees under extreme or adversarial conditions.
> > >
> > > **2\.** We appreciate the reviewer’s emphasis on computational practicality. We note that our initial memory reporting in response to Reviewer BD7b overstated the computational burden of our framework as it *included both the base-model footprint and the Jacobian computation.* We have also: 1) optimized the offline Jacobian pipeline using a *Jacobian Vector Product (JVP) framework*, rather than autograd, which is exact but does not require storing the backwards computational graph on GPU memory; and 2) *optimized our implementation (e.g., in-place operations)*.
> > >
> > > Through these changes we substantially reduce the memory footprint, and make our approach feasible for practical hardware limitations. **Notably, the full pipeline for Qwen-2.5-14B now runs on a 4090 GPU, and the 32B model is reduced to less than 40GB (compatible with an A100 GPU).** With the optimizations included, a more detailed breakdown of memory is presented below:
> > >
> > > ||Model|Jac. (GB)|Model (GB)|
> > > |---|---|---|---|
> > > |Fast|Qwen14B|12.99|26.06|
> > > |Fast|Qwen32B|17.25|38.73|
> > > |JVP|Qwen14B|3.98|13.617|
> > > |JVP|Qwen32B|6.33|20.798|
> > > |JVP|Llama70B|28.32|40.721|
> > >
> > > As observed in the table, the memory requirements of our method are modest relative to the base model footprint (e.g., 30% increase for Qwen models with JVP). This is especially apparent and valuable for smaller models <= 14B, most typical in activation steering literature [1]. *For larger models, the base model itself exceeds many workstation limits, but our method remains modest in comparison.* Furthermore, the Jacobian computation and Riccati recursions occur offline, and the inference-time memory footprint of our method only requires storing the gain matrix K, *resulting in an even smaller memory footprint during inference time:*
> > >
> > > | Model | Base Mode (GB)| K (GB) | Low-Rank K (rank=500) (GB) |
> > > |---------|------------|---------|---------|
> > > | Llama8b | 9.647 | 1.989 | 0.477 |
> > > | Qwen14b | 17.289 | 4.678 | 0.906 |
> > > | Qwen32b | 27.903 | 6.235 | 1.205 |
> > >
> > > Finally, we provide empirical motivation for a path forward via evaluation of a low-rank approximation of the gain matrix K. While a true low-rank pipeline including low rank Jacobians and Riccati (discussed in [2]) is beyond the scope of this work, we evaluate a low-rank K matrix derived from a precomputed full Jacobian, reducing the online footprint (e.g., reduces to 1GB from 6GB for Qwen-2.5-32B – summarized above). As seen in the table below, low-rank K can achieve competitive steering performance with the full rank matrix – full evaluations are provided in the supplemental materials.
> > >
> > > |Model|Rank|Tox (%)| PPL|
> > > |---|---|---|---|
> > > |gemma2b|2304|0|12.040065|
> > > ||10|1.6|10.615625|
> > > ||500|0.2|11.812355|
> > > |llama8b|4096|0|10.456625|
> > > ||10|3.4|7.222910|
> > > ||500|0|11.181273|
> > >
> > > [1] Zhao et. al., “ODESteer…,” ICLR, 2026
> > >
> > > [2] M. Cho et. al., “Low-Rank LQR Optimal Control Design…,” ECC 2023

---

### Official Review · Reviewer_YJUu · 2026-03-12

**Soundness:** 2
**Presentation:** 3
**Significance:** 2
**Originality:** 3
**Overall Recommendation:** 4
**Confidence:** 3

**Summary:**

This paper provides a baseline for analysing local linearity in the LLM activation space, enabling application of an LQR controller to steer activations toward an array of linear feature setpoints. The local linearity of LLMs is experimentally supported by the cosine similarity of Jacobian matrices and theoretically bounded modelling error. Given the local linearity, a linear time-varying state-space representation of LLM activation dynamics is mathematically provided. Considering activation steering as a linear control problem, the paper establishes an A-LQR framework to optimise the steering process towards feature-strength specific setpoints. Experimental results across different models demonstrate significant improvement on toxicity mitigation and truthfulness elicitation compared to other methods.

**Compliance With Llm Reviewing Policy:**

Affirmed.

**Final Justification:**

Although the empirical evidence supporting the local linearity of the Jacobian matrix is not fully persuasive, and there remains ambiguity regarding the actual need to use LQR to stabilise performance gains (since the experiments suggest this can be achieved by scaling the $\lambda$ value in LFS), I still find the overall idea and execution of this paper to be novel and interesting. Hopefully it will inspire more ideas for applying control theory in activation steering. I am leaning toward acceptance, though I would not mind rejection.

**Key Questions For Authors:**

In addition to the questions raised in the weaknesses above:

1. It is unclear how different $Q$ and $R$ matrices affect steering performance. In the D.1 section, the paper argues that increasing or decreasing $Q$ yields different feature strengths. However, the feature strength control could also be achieved by the $\lambda$ parameter. Consequently, the role of adopting LQR control for steering is questionable. Could the authors clarify what additional degrees of freedom the Riccati recursion offers beyond $\lambda$ scaling?

**Limitations:**

- There is no specific section declaring the limitations of the approach.
- The paper lacks detailed ablation studies on scenarios where A-LQR might underperform or destabilise.

**Strengths And Weaknesses:**

## Strengths

**Soundness**
- The paper includes a comprehensive analysis of the LLM local linearity by mathematical formalisation and experimental observation.
- Experiments show significant improvement in steering toward the target behaviour while maintaining model performance.

**Originality**
- It is novel to exploit the local linearity within the LLM's activation space to apply LQR in activation steering.

## Weaknesses

**Soundness**
- There is no comparison between LFS and other baselines in the setpoint selection for steering. All experiments bundle LFS and A-LQR together (Tab. 1, 2), so it is unclear whether gains stem from the control framework or the setpoint construction. Running LFS with simpler methods (e.g., direct addition) or external baselines' directions with A-LQR would isolate the respective contributions.
- The layer-wise Jacobian coupling results show moderate similarity values (mostly ~0.6–0.7) in intermediate layers (Fig. 5). The authors do not establish what similarity level is required for the controller to remain effective, nor analyse per-layer controller quality as a function of alignment. Without this, the evidence for the linearity assumption in those layers is insufficient.
- The performance gap over external baselines is very large (e.g., Llama-3-8B: A-LQR 0.12% vs ITI 0.64% in Tab. 1). Given that LFS uses a standard mean-difference construction, it is unclear what accounts for this margin without proper ablations isolating the contribution of LFS versus the LQR controller.

**Presentation**
- Typo in the caption of Figure 5: "Gemma-2-B" -> "Gemma-2-2B"
- The definition of $Q_f$ in Section D.1 could not be located.

**Significance**
- The computational cost warrants further scrutiny. The authors claim the Jacobians can be reused across prompts (since they are locally similar), but computing Jacobians for each layer is still $\mathcal{O}(\ell d^3)$ and memory-intensive. For the small models here (1B–14B) this is manageable, but scalability to larger models is not addressed.
- No limitations section or analysis of failure modes.

**Originality**
- The local linearity finding, while presented as a core contribution, has been partially observed in prior work on Jacobian structure [1]. The contribution is more in exploiting this structure for control synthesis than in discovering it.

---

> ### Author Rebuttal · Authors · 2026-03-31
>
> We sincerely thank the reviewer for the insightful feedback. We have addressed all points and welcome further questions.
> [Supplemental](https://anonymous.4open.science/r/paper_supp-E476/_ICML_2026__Rebuttal_Figures_and_Tables_sanitized.pdf )
>
> ### **Q2.1: Source of performance gains - control vs setpoint**
> We note Sec. 7.1 includes S-PID as an ablation on the LQR control strategy by tracking the same LFS setpoint via PID control (lines 361-363R) [1]. We will revise the text to clarify that S-PID is an ablation for isolating effectiveness of LFS. Inspecting Tables 1-2, A-LQR reliably outperforms S-PID across models on both toxicity mitigation and truthfulness, though S-PID outperforms baselines from literature.
>
> Notably, S-PID reduces to ActAdd with LFS when the integral and derivative gains are set to 0. To strengthen this comparison, we implement an explicit ActAdd baseline tracking LFS, ablating over intervention layers. Complete results provided in supplemental:
>
> | |Method|CLS Tox. (%) (↓)|PPL (↓)|
> |---|---|---|---|
> |Gemma-2-2B|Original|4.16 ± 0.54|8.95 ± 0.07|
> ||ActAddLFS|0.82 ± 0.20|9.90 ± 0.08|
> ||A-LQR|0.18 ± 0.08|12.26 ± 0.08|
>
> ActAddLFS and S-PID show that LFS is useful, but the control strategy is critical: only A-LQR consistently achieves strong, stable steering. We will revise the paper to make this decomposition explicit and include ActAddLFS results.
>
> [1] Nguyen et. al., “Activation steering with a feedback controller.” arXiv:2510.04309
>
> ### **Q2.2: Controller Quality as a Function of Alignment**
> We agree that establishing a quantitative link between Jacobian alignment and controller effectiveness strengthens the case for approximate linearity in intermediate layers. We clarify that the similarities reported in Fig. 5 are ~0.3–0.5 on average depending on model (rather than ~0.6–0.7).
>
> We quantify the link between Jacobian alignment and controller efficacy by progressively corrupting layer-wise Jacobians with random noise, measuring the impact on concept steering. This identifies per-layer alignment thresholds corresponding to a fixed performance cutoff (e.g., concept prevalence = 0.7).
> Representative results for Llama-3-8B are shown below. Observed mean alignments for layers 0, 8, 16, and 24 (0.32–0.41) exceed these thresholds, indicating that alignment is sufficient for effective steering, despite stochastic variability. More results in the supplemental.
>
> |Layers|Threshold Band|
> |---|---|
> |(24)|[0.242,0.250]|
> |(1,15)|[0.287,0.291]|
> |(4,5,9)|[0.286,0.312]|
>
> ### **Q2.3: Performance gap over external baselines**
> Our ablations above (ActAddLFS, S-PID) show that LFS alone does not achieve strong or consistent performance, while A-LQR robustly tracks the setpoint under model dynamics. We will make this distinction explicit in the revision.
>
> ### **Q2.4: The computational cost warrants further scrutiny**
> We provide benchmarking (see Reviewer ApYn), including larger models (32B, 70B). Jacobians can be reused across prompts and tasks, but computing them remains costly, particularly for large models. Future work should explore optimizations such as low-rank approximations of Jacobians or stored feedback matrices.
>
> ### **Q2.5: No limitations section or analysis of failure modes.**
> We will add a dedicated section. Computing layer-wise Jacobians can be memory-intensive, particularly for medium or large models. While memory optimizations are possible at the cost of runtime, this remains a potential bottleneck. Preliminary experiments suggest low-rank approximations of Jacobians may be effective, though a full evaluation is beyond the scope of this work.
>
> Furthermore, our method assumes approximate local linearity, which empirical results (Sections 5–7) suggest holds in practice, but this assumption may be sensitive to adversarial or edge-case inputs. That said, the linear formulation enables the application of robust control techniques to explicitly address such challenges, providing a foundation for future research in linear control-based steering.
>
> Finally, the LQR-based framework is sensitive to the Q and R gains, which can require extensive tuning.
>
> ### **Q2.6: Local linearity finding partially observed in prior work [1]**
> We could not identify the specific reference [1]. To the best of our knowledge, our work is the first to characterize layer-wise Jacobian coupling. While related inter-layer Jacobian structures have been discussed in prior work [2], our contribution highlights intra-layer coupling which enabling LTV control synthesis.
>
> [2] Aubry et. al., “Transformer Block Coupling…” ICLR, 2025
>
> ### **Q2.7: Clarify what flexibility LQR offers beyond λ scaling**
>
> To clarify, λ defines the target setpoint, while Q and R govern the trade-off between tracking error and control effort for achieving that target setpoint. The Riccati recursion then computes optimal gains under these trade-offs. We include ablations on how Q and R affect steering performance for a fixed λ (Fig. 15) in the submitted appendix.

---

> > ### Author Rebuttal · Reviewer_YJUu · 2026-04-04
> >
> > I thank the authors for their thorough rebuttal and the additional experiments provided. I have the following follow-up questions:
> >
> > 1. Given the simplicity of the ActAddLFS framework, its performance gains are unexpectedly large, such as the CLS Tox score for Gemma-2-2B dropping from 4.16% to 0.82%. Could the authors please justify why this naive baseline achieves such substantial improvements? Furthermore, if A-LQR is intended to stabilize generation, why does it consistently introduce higher perplexity compared to the simpler ActAddLFS baseline?
> > 2. The corrected Jacobian similarity values (~0.3-0.5) are lower than I originally reported. Could the authors comment on why these lower values are still sufficient for the linearity assumption, beyond the task-specific threshold analysis provided ?
> > 3. The performance cutoffs used to derive the alignment thresholds differ across models (0.7 for Qwen-2.5-3B, 0.88 for Llama-3-8B, 0.6 for Gemma-2-2B in supplemental Tables 2-4) without justification for these choices. If these cutoffs are selected post-hoc per model, it is unclear how meaningful the resulting thresholds are. More broadly, how do these thresholds generalise to other steering objectives such as toxicity mitigation ?
> > 4. I apologise for the citation confusion. The reference [1] in my review was indeed Aubry et al., which the authors correctly identified. However, having reviewed Aubry et al. more carefully, I note that their work does examine coupling across tokens at the same layer (not only across depth). I accept that the focus on stability across different reachable activations for control synthesis is distinct, but would suggest the authors acknowledge this nuance more carefully in the revision rather than drawing a sharp inter-layer vs intra-layer distinction.

---

> > > ### Author Response · Authors · 2026-04-08
> > >
> > > We thank the reviewer for their continued feedback. We address the followup questions below: ([Supplemental](https://anonymous.4open.science/r/paper_supp-E476/reply.pdf))
> > >
> > > **1\.** We agree that the ActAddLFS result is strong and does not contradict our claim that both LFS and LQR matter for effective steering. Its performance aligns with prior work showing that simple steering methods can already yield large gains: on Gemma-2-2B, ActAdd reduces toxicity from 4.2% to 1.1%, so ActAddLFS at 0.82% is a modest improvement. S-PID, which also uses the LFS signal, performs similarly at 0.86%. A-LQR further shows that, once the LFS target is fixed, modeling layerwise dynamics improves the tradeoff between steering strength and output quality, achieving 0.12–0.18% toxicity while maintaining reasonable PPL across models.
> > >
> > > More specifically, A-LQR minimizes control effort while tracking a target setpoint, so it may accept higher PPL when enforcing stronger targets (see manuscript, lines 407-425L). PPL should therefore be interpreted jointly with achieved toxicity.
> > >
> > > When baselines match A-LQR on toxicity, they often do so with a larger PPL spike, see ODESteer in Supplement Table 9. Baselines that perform well on one model also often fail to generalize across models, whereas A-LQR is more consistent. Appendix D.2 shows that the advantage of A-LQR is more apparent at stronger target setpoints, where it preserves coherence better than PID-style feedback. We also include another A-LQR configuration that achieves greater toxicity reduction than ActAddLFS with comparable PPL:
> > >
> > > |Model|Method|Tox. (%) (↓)|PPL (↓)|
> > > |-|-|-|-|
> > > |Gemma-2-2B|Original|4.16 ± 0.54|8.95 ± 0.07|
> > > ||ActAdd|1.10 ± 1.80|11.42 ± 0.58|
> > > ||ActAddLFS|0.82 ± 0.20|9.90 ± 0.08|
> > > ||A-LQR*|0.34 ± 0.01|10.15 ± 0.08|
> > >
> > > We will revise the text to avoid calling A-LQR “stable” in an absolute sense, and instead clarify that it offers a more favorable tradeoff between steering performance and auxiliary metrics (e.g., PPL).
> > >
> > > **2\.** We agree that values in this range should not be described as “near equality” of Jacobians, and we will revise the paper to make this explicit. Rather, they indicate sufficient alignment in dominant Jacobian subspaces for effective closed-loop control. For high-dimensional, fully-actuated LLM dynamics (i.e., we can independently modify every latent dimension), a locally accurate model suffices for feedback control, as the controller can compensate for errors arising from imperfect approximations. Indeed, the supplemental (Thm. 1) shows that reducing actuation worsens tracking bounds and performance.
> > >
> > > Hence, the alignment metric should not be viewed in isolation, but together with the observed low tracking error. In particular, Thm. 4.2 bounds how controller tracking performance degrades as the model error bound increases. In Sec. 5 and appendix, closed-loop tracking error is bounded and the worst-case bound contracts across models. Thus, we do not present the alignment values as proof of exact linearity, but as evidence that the local approximation is *informative enough* (through theoretical analysis in Thm. 4.2 and Figs 3,10,11; being above identified performance thresholds; and through empirical success in Sec. 6-7) for the controller to work in practice.
> > >
> > > **3\.** We agree that this selection procedure should be made explicit. These cutoffs are not tuned post-hoc per model. Rather, for each model we select the performance cutoff near the mean performance under Jacobian corruption, to ensure sufficient data on either side of the cutoff to reliably estimate the alignment thresholds. While the exact values vary across models due to differing base performance distributions, the selection procedure is consistent. Details will be included in the revision.
> > >
> > > We do not claim a universal threshold, as it is intrinsically task and parameter dependent: changing both the steering objective and LQR/LFS parameters changes the definition of “sufficient” performance. Thus, the key result is that the alignment can inform steering performance, and that the observed alignment between Jacobians generally lie above the empirically determined threshold.
> > >
> > > Additionally, we observe similar behavior for toxicity mitigation, with a representative example below. While the specific performance cutoffs must be defined per task (see supplemental), the key result remains consistent.
> > >
> > > |Layers|Threshold Band|
> > > |-|-|
> > > |(5,16,24)|[0.192,0.192]|
> > > |(22,15,33)|[0.189,0.190]|
> > > |(12,13,27,5)|[0.189,0.199]|
> > >
> > > **4\.** Thank you for the clarification. We agree that our earlier wording overstated the distinction, and our contribution is narrower: we study the similarity of fixed-layer Jacobians across reachable activations, and show that this enables reuse of offline-computed feedback gains for LTV controller synthesis. We will revise the text to acknowledge this overlap and frame our contribution as complementary rather than sharply contrasting with Aubry et al.

---

### Official Review · Reviewer_BD7b · 2026-03-12

**Soundness:** 3
**Presentation:** 3
**Significance:** 4
**Originality:** 4
**Overall Recommendation:** 5
**Confidence:** 2

**Summary:**

The paper studies activation steering for LLMs, i.e. applying inference-time interventions to activations that result in some desired behaviour. The authors do so by treating transformer blocks as locally linear and allowing this way a control theoretic solution to the problem. The latter first introduces layerwise feature setpoints and then linearizes the model around a trajectory to derive a closed-loop LQR controller that steers hidden states during generation. The broad empirical evidence is promising and spans OneSeC for concept induction, RealToxicityPrompts for toxicity, TruthfulQA generation and MMLU for truthfulness/generalization, evaluated across 6 models from the Llama, Gemma, and Qwen families.

**Compliance With Llm Reviewing Policy:**

Affirmed.

**Final Justification:**

A well written and easy to follow paper stemming from an intuitive motivation. Analyses are well thought and convincing and the results support the method.
The rebuttal addressed my already few concerns which mostly revolved around minor issues and lack of runtime discussion. I am confirming my accept decision.

**Key Questions For Authors:**

- Why does the approach fail for Qwen-2.5-3B? the paper claims it is an exception but does not provide further discussion or analysis.
- When is local linearity expected to break down? is it related to the method failing over Qwen?

**Limitations:**

yes

**Strengths And Weaknesses:**

### Strengths

- The paper is well written and easy to follow. The motivation makes intuitive sense, with the introduction outlining practical limitations of prior activation steering techniques and the proposed approach elegantly solving these ones. Analyses are well thought and convincing.
- The approach is interesting and smart. Contrarily to a large fraction of more heuristic activation steering works, framing activation steering as a model-based control problem is a reasonably principled approach.
- The local-linearity analysis is particularly interesting. I find the Jacobian analysis to be one of the more compelling parts of the paper and also potentially relevant beyond mechanistic interpretability.
- The empirical evidence is solid and broad. The method is evaluated on arbitrary concept induction, toxicity mitigation, and truthfulness across six open-source models from the Llama, Gemma, and Qwen families against a fairly large set of baselines. The results strongly support the approach.

### Weaknesses
- The efficiency claim should be supported by runtimes. The paper says the controller has minimal computational overhead and argues that offline gains can be reused but there is no actual runtime, latency or memory benchmark against the baselines. Given how lightweight activation steering methods usually are, this might be one potential disadvantage of the method that should be known.
- As a minor point, the notation is somewhat heavy and visually overcrowded in some places. In particular, the frequent use of layer, time, and feature indices makes some equations harder to parse than they could otherwise be.
- Again minor, but the manuscript should be proof-read as I’ve found several typos, e.g.  L055: a inference time paradigm, L167: activtion, L811: inlcuded.

---

> ### Author Rebuttal · Authors · 2026-03-31
>
> We sincerely thank the reviewer for their thoughtful comments, and we carefully address the suggestions you raised below. Please let us know if you have any additional questions.
>
> ### **Q1.1: The efficiency claim should be supported by runtimes and memory benchmarks**
>
> We summarize the online runtime requirements in response to Reviewer ApYn.
>
> For offline computation, the primary overhead is the computation of layer-wise Jacobians. This is a memory intensive task, with VRAM dependency quadratic in the model's latent dimension $d$. This is manageable for 1B-14B models, but approaches practical hardware limits for larger models. We perform evaluations on two practical approaches for computing the model Jacobians: (1) directly computing full Jacobians with vectorized operations, which is faster but approaches hardware limitations for VRAM requirements (denoted "Fast" in the table); and (2) computing Jacobians row-by-row, which is substantially slower, but substantially reduces the memory footprint (denoted "VRAM-efficient" in the table). All evaluations are conducted on a single H100 GPU, with the exception of the Llama-3.1-70B which was evaluated on a H200.
>
> This offline computational burden is certainly a limitation of the approach, and future work would involve exploring different optimization strategies, and low-rank approaches. We discuss this further in our proposed Limitations section, discussed in the response to Reviewer YJUu.
>
>
> |Mode|Metric|Llama-3.1-70B|Qwen2.5-32B|Qwen2.5-14B|Llama-3-8B|Gemma-2-2B|
> |---|---|---|---|---|---|---|
> |**Fast**|Runtime(s)|344.54 ± 7.73|149.25 ± 21.41|74.10 ± 1.90|34.58 ± 0.50|10.96 ± 0.29|
> ||VRAM(GB)|124.53|78.44|66.23|38.02|15.28|
> |**VRAM-efficient**|Runtime(s)|1283.20 ± 2.40|892.80 ± 3.13|449.53 ± 3.15|242.07 ± 0.89|117.61 ± 0.21|
> ||VRAM(GB)|89.83|54.13|35.05|17.35|5.00|
>
>
> ### **Q1.2/3: Notation and Presentation:**
>
> Thank you for pointing out unclear notation and typos. In the final submission, we will aim to reduce the notation, omitting indices when the context is clear.
>
>
> ### **Q1.4: Why does the approach fail for Qwen-2.5-3B?**
> We thank the reviewer for highlighting this and agree that the behavior on Qwen-2.5-3B merits further discussion.
>
> On toxicity mitigation, A-LQR (ours) achieves similar reductions as on other models, with MMLU performance remaining consistent. The main deviation is a sharp increase in perplexity (PPL), which is not observed on other models or tasks.
>
> A plausible explanation is that Qwen-2.5-3B is unusually sensitive to A-LQR perturbations, causing disproportionate changes in PPL despite stable high-level behavior. This effect does not appear in other small models (e.g., Gemma-2-2B) or in Qwen-2.5-14B, suggesting it is specific to this model rather than scale or architecture alone.
>
> We therefore consider this an outlier, not a general limitation of the method, and will revise the appendix to highlight this and include further analysis (e.g., hyperparameter sensitivity).
>
>
> ### **Q1.5 When is local linearity expected to break down?**
>
> Local linearity is expected to break down when either the activations move far from the linearization point, or the model output changes rapidly in its neighborhood, making a linear approximation inaccurate. In such cases, nonlinearities dominate and the local model no longer captures the true dynamics. This is more likely under inputs that induce large embedding shifts (e.g., some form of adversarial prompts). However, our experiments (Section 5) suggest that for natural, task-relevant prompts, such shifts are rare (see the response to Reviewer YJUu).
>
> When local linearity fails, the controller must compensate for larger model mismatch, requiring greater control effort or failing to track the reference, which can degrade both the core steering success and auxiliary performance such as PPL.
>
> We do not believe this explains the behavior on Qwen-2.5-3B. As noted above, that case shows stable task performance but elevated PPL, suggesting model-specific sensitivity to perturbations rather than a failure of the linear approximation.
>
> We will clarify this distinction in the revision and expand discussion of when local linearity holds, including additional analysis relating failure modes to our proposed Jacobian alignment metric, which we outline in response to Reviewer YJUu.

---

> > ### Author Rebuttal · Reviewer_BD7b · 2026-04-01
> >
> > I thank the authors for their rebuttal.
> > I am glad to see my few concerns addressed. I am confirming my positive score.

---

> > > ### Author Response · Authors · 2026-04-08
> > >
> > > We sincerely thank the reviewer for their thoughtful review and for acknowledging our revisions. We greatly appreciate the positive feedback and are glad we were able to address their concerns.

---

### Decision · Program_Chairs · 2026-04-30

**Decision:**

Accept (regular)

**Comment:**

This paper introduces a test-time activation steering method for LLMs, grounded in a theoretical framework based on the local linearity of transformer activations and formulated from a model-based optimal control perspective. The reviewers' opinions initially diverged, but the discussion during the rebuttal and AC-reviewer phase led to a convergence toward acceptance (while still including weak'' accepts). All reviewers seem to agree that the core idea, exploiting local linearity in LLM activations, is novel and interesting, and that the theoretical formulation is reasonably well-motivated. Furthermore, the paper provides extensive experiments across multiple LLMs, offering solid empirical support for the proposed approach.

However, at the same time, several concerns remain. In particular, computational cost is identified as a major limitation, especially for very large models (BD7b, YJUu, abn9, ApYn). While the authors acknowledge this limitation, the AC encourages them to more clearly articulate and discuss this constraint in the final version. Additionally, reviewers (YJUu, abn9) note that the empirical evidence supporting the local linearity assumption, particularly regarding the Jacobian, would benefit from more thorough analysis and clearer exposition. Strengthening this aspect would further solidify the paper's contributions.

Overall, the strengths, particularly the method's novel perspective, solid theoretical grounding, and extensive empirical validation, outweigh the identified weaknesses. Therefore, the AC recommends acceptance. This decision is conditional on the authors addressing the above concerns in the final revision.